# Association of excessive screen time exposure with ocular changes leading to astigmatism in children

**Mutahir Shah**[1]*, **Satheesh Babu Natarajan**[1], **Nafees Ahmad**[2]

**1** Department of Applied Sciences, Sub Division Health Sciences, Lincoln University College, Pata Lang Jaya, Selangor, Malaysia, **2** Institute of Biomedical and Genetic Engineering, Islamabad, Pakistan.

* mshah@lincoln.edu.my

## Abstract

### Purpose

The purpose of this study was to find the relationship between excessive screen time exposure and the development of astigmatism in children.

### Methods

A cross-sectional correlational study was conducted at tertiary care hospital, Islamabad between June 2023 to May 2024. In total, 431 patients were included in this study after informed written consent. Screen time exposure was assessed through smartphone usage history or parental reports of daily use of smart gadget(s). Tear break-up time was measured using a fluorescein strip, cobalt blue filter on a slit-lamp biomicroscope and a stopwatch. Data was analyzed using univariate and multivariate statistical tests including Spearman's correlation and regression analyses.

### Results

The study included 431 children (mean age: 6.70 ± 1.80 years; 55% male, 45% female). Mean screen time was 4.54 ± 1.52 hours/day. A positive correlation between screen time and the magnitude of astigmatism was observed (r = 0.33, p < 0.001). Regression analysis showed a significant relationship among screen time and astigmatism (B = 0.177, CI: 0.80-0.25). Tear break-up time (TBUT) showed a significant negative correlation with screen time (r = -0.167, p < 0.001), and reduced TBUT was linked to a higher risk of inflammatory conjunctivitis and lid thickening (B = -0.431, CI: -0.12 to -0.49, p < 0.001). The results highlighted that inflammatory conjunctivitis/lid thickness have three times greater risk of developing high astigmatism (OR = 3.31, p-value < 0.001, CI = 1.91 to 5.73) while the risk of moderate astigmatism in such cases was two times higher (2.12, p-value = 0.004, CI = 1.26-3.56). However, the effect of lid thickness on astigmatism when combining with screen time has a little effect that is not significant (p-value = 0.053). Thus, excessive screen time is an independent risk factor of causing astigmatism in children (p < 0.001).

**Data availability statement:** All relevant data are within the paper and its Supporting Information files.

**Funding:** The author(s) received no specific funding for this work.

**Competing interests:** The authors have declared that no competing interests exist

The Wilcoxon Signed Ranks Test demonstrated significant improvement in visual acuity after correction ($p < 0.001$).

## Conclusion

Excessive screen time in children is significantly associated with astigmatism, tear film instability, inflammatory ocular conditions, including conjunctivitis and lid thickening. These findings suggest the need for preventive strategies, such as reducing screen time and encouraging regular eye examinations, to protect children's ocular health.

## Introduction

Emerging technologies, such as smart screen gadgets and interactive screen media, have become an integral part of youth and children's daily life. These technologies are now widespread and significantly influence our daily lives, including the habits of youngsters and children [1,2]. Today's children are genuinely "digital natives" in a constantly evolving digital world. The average age at which children begin interacting with media has decreased from 4 years to 4 months since 1970 [3]. While digital devices offer educational and recreational benefits, excessive screen time exposure has raised significant concerns regarding ocular and general health. The potential detrimental effects on children's eye health and development have prompted worries about the alarming increase in screen usage, as evidenced by multiple studies [1,2,4]. Most pediatric associations worldwide recommended that children should limit their recreational screen exposure to less than 2 hours per day in response to these concerns [3,5].

Exposure to excessive screen time during early life is linked with refractive errors. Uncorrected refractive errors are responsible for 43% of global visual impairment [6,7]. They were further classified into myopia, hypermetropia and astigmatism. Myopia is the commonest refractive error that required correction in early age. Its causes, risk factors and management options have been extensively reported [8]. While astigmatism is relatively less studied among refractive errors. It became more prevalent worldwide in recent years, raising important clinical and public health concerns [9]. If left untreated, astigmatism significantly reduces visual acuity, negatively affects child's visual development, and may result in amblyopia [10,11].

Apart from that, studies showed that prolonged exposure to screens in children is associated with abnormal ocular surface and dry eyes [12–14]. Excessive screen use is thought to be a contributing factor to dry eye diseases (DED), as it alters blink patterns, leading to an abnormal ocular surface. Decreased spontaneous blinking and incomplete eyelid closure during prolonged screen time affect tear film stability, leading to increased aqueous evaporation, altered osmolarity, and the buildup of inflammatory mediators [15]. These inflammatory mediators are proposed to be associated with an ocular inflammatory response. This ocular inflammatory response is the key to inflammatory conjunctivitis that is proposed to cause lid thickening. This mechanism may cause mechanical reshaping of the cornea. Previous studies had shown that, in children and teenagers, factors such as extraocular muscle tone, excessive screen time, eyelid pressure, heredity, visual input, and environmental pollution may play a role in the pathogenesis of astigmatism, although the exact pathogenesis remains unclear [9,10,16,17]. Thus, there is a gap in the current literature regarding direct or indirect association of screen time with the development of astigmatism. Identifying modifiable risk factors that may contribute to the development of astigmatism during the critical period is therefore essential.

The aim of this study was to investigate the relationship between excessive screen time exposure and the development of astigmatism in the pediatric population. Furthermore, the study highlighted the effect of thickened upper lid due to dry eyes on development of astigmatism.

## Methodology

This cross-sectional correlational study was conducted in the department of ophthalmology at tertiary care hospital, Islamabad. Ethical approval was granted by the hospital's ethical review committee ref no:XXX-HI-PUB-ERC/Mar23/24. All steps were followed as per the guidelines of the Declaration of Helsinki. Informed written consent was obtained from the parents/guardians. Additional information regarding the ethical, cultural, and scientific considerations specific to inclusivity in global research is included in the supporting information (Inclusivity-in-global-research-questionnaire). In this study, 431 individuals aged 3 to 11 years were included through a consecutive sampling technique (non-probability sampling). The sample size was calculated using WHO calculator with a 95% CI and considering pediatric astigmatism prevalence of 14.9% (0.149) [1]. The study was conducted from June 2023 to May 2024. Participants with headaches, squint, unhabitual blinking, meridional amblyopia, astigmatism ≥ -0.75, and a history of excessive gadget use, i.e., smartphones, tab and laptop (≥ 2 hours/day for ≥ 6 months) were included. Exclusion criteria included children with retinal dystrophies and diseases, trauma, ectasias, keratoconus (following Rabinowitz criteria), Ehlers-Danlos syndrome, Down syndrome, Marfan syndrome, vernal keratoconjunctivitis, atopy, history of contact lens use, history of squint surgery, maternal age > 35 years [2], birth weight < 1500g [3], gestational period < 37 weeks [2] and history of ocular surgeries or any systemic illness. Exposure to TV screens was excluded from the study because recent research predominantly associates it with myopia rather than astigmatism [4]. Screen time was evaluated using smartphone usage history, supported by parental reports for children with dedicated gadgets. For non-dedicated gadgets, parental reports on their children's usage were utilized (S1 Fig. The figure highlights screen time usage among children with dedicated devices).

Visual acuity (uncorrected and corrected) was measured using a log MAR visual acuity chart held at a distance of 4 meters. Refraction was performed with an autorefractometer and retinoscopy using (Nidek ARK-1, Japan) and retinoscope (NIETZ, Japan). Cycloplegic refraction was performed after instilling 1% cyclopentolate drops three times, with a duration of 10 minutes between each instillation, and the child was re-examined after 60 minutes [18]. Corneal curvature was measured using an OCT-based corneal topography instrument (OPTOPOLE Revo NX-130, Poland).

Tear break-up time was documented using fluorescein strip instillation, followed by evaluation with a cobalt blue filter on a slit-lamp biomicroscope and a stopwatch. Conjunctival tissue was assessed using a slit-lamp biomicroscope for inflammatory response and papillary reactions. The inflammatory conjunctivitis was supposed to be associated with thickening of the tarsal plat of the conjunctiva [2]. The thickening or swelling of the conjunctival tarsal plate was clinically assessed on slit lamp examination based on conjunctival inflammation and digital assessment for overall lid rigidity and thickness. Blink rate while staring at a smartphone was measured using a video analyzer. Ocular deviation was measured using loose prisms and alternate cover test.

Data were analyzed using SPSS version 26. Descriptive statistics were applied for continuous variables (e.g., age, visual acuity, astigmatism magnitude), while categorical variables were graphically presented. The data did not follow a normal distribution according to the Shapiro-Wilk test criteria and was further confirmed by the normality assessment using histogram

plots (S2 Fig. The figure showed the graphical presentation of the histogram plots highlighting the normality of the data). Spearman's correlation analysis was performed to find the association between the magnitude of astigmatism and screen time. Univariate analysis was conducted among dependent and independent variables along with gender and family history. In multivariable linear regression model assessed the relationship between screen time and astigmatism, TBUT, and conjunctival inflammation. Bonferroni adjustment for multiple comparisons was utilized to establish the p-value for the multivariable analysis which was calculated to be 0.025. Chi-square tests, binary, and multinomial logistic regression analyzed associations between TBUT, astigmatism, lid thickness, and conjunctivitis. Wilcoxon Signed-Rank test was used for visual acuity comparison. A p-value of ≤ 0.05 was considered as statistically significant (SPSS data).

## Results

In total 431 patients were included in this study. The study population consisted of children with a mean age of 6.70 ± 1.80 years. Both genders were included in the study, with 55% males and 45% females. The feeding patterns of the children were breastfed (69%), followed by bottle-fed (15.5%) and mixed feeding (15.5%). Birth history showed a balanced distribution between vaginal delivery (VD; 51.7%) and caesarean section delivery (CSD; 48.3%). A family history of refractive error (myopia, hyperopia, astigmatism) was positive in 25.5% of the participants. Parents were educated to graduate or postgraduate levels (S3 Fig. This figure showed parents educational level of the study participants. It showed that mostly they were educated to graduate and post graduate level.), with incomes typically falling within tax slabs 3 (2.2-3.1 million per annum) and 4 (3.1-4.2 million per annum) designated by the government of Pakistan (S4 Fig. Presented the status of parental income per annum as per Govt of Pakistan tax slab status, majority were lying in tax slab 3 and 4). Intermittent exotropia was common among the study population. Collinearity for all variables between the right and left eyes was highly significant (p-value < 0.001), so only the results of the right eye were included. The average uncorrected visual acuity was 0.59 ± 0.18 LogMAR (0.511 Log MAR in hormonic means value), which improved with spectacle correction to a mean of 0.18 ± 0.14 LogMAR. The mean screen time exposure was 4.54 ± 1.52 hours per day, and the average outdoor activity duration was 30.6 minutes. The mean total and corneal astigmatism were -2.44 ± 1.15DC and -2.38 ± 0.94DC, respectively (Table 1). Astigmatism severity was categorized into three groups: mild (≥-0.75 to -1.50DC), moderate (>-1.50 to -2.50DC) and high (>-2.50DC). Group 1 consisted of 87 participants, while group 2 had 182 and group 3 had 162 participants (Fig 1). The data also included tear break-up time (10.34 ± 3.19 seconds) and blink rate (10.07 ± 3.03 blinks per minute; Table 1). The mean corneal thickness was 542 ± 32.43μm (480-610). All study participants showed astigmatism "with the rule" and it was mostly corneal in nature. Mixed astigmatism was the most prevalent type of astigmatism (47.8%), followed by compound hyperopic astigmatism (19.3%), simple myopic astigmatism (16.7%), compound myopic astigmatism (11.6%) and the least common was simple hypermetropic astigmatism (4.6%; S5 Fig. Represented the types of Astigmatism on the basis of Foci, it showed that majority of the study participants were diagnosed with Mixed Astigmatism followed by compound hyperopic, Myopic and simple Hyperopic Astigmatism).

Symptoms of inflammatory conjunctival tissue were common in approximately 59.80% of participants, correlating with a thick tarsal conjunctival plate. The mean value of tear film stability, as measured by tear break-up time, was 10.80 ± 3.25 seconds (Table 1). The data revealed that 48.5% of children exposed to gadgets were diagnosed with abnormal tear film. The classification of tear break-up time showed mild TBUT in 31.1%, moderate in 16.5%, and

**Table 1.** Descriptive analysis of demographic and clinical parameters (n = 431).

| Demographics and Clinical Data | Mean ± SD | Minimum | Maximum |
|---|---|---|---|
| Age | 6.67 ± 1.90 | 3 | 11 |
| Birth Weight | 3.01 ± 0.34 | 2.00 | 4.20 |
| Gestational Period (weeks) | 38.75 ± 1.36 | 37 | 42 |
| Visual Acuity Uncorrected (LogMAR) | 0.59 ± 0.18 | 1.20 | 0.10 |
| Visual Acuity Corrected (LogMAR) | 0.18 ± 0.14 | 0.00 | 0.50 |
| Corneal Astigmatism (DC) | -2.38 ± 0.94 | -0.75 | -6.0 |
| Magnitude of Astigmatism (Total, DC) | -2.44 ± 1.15 | -0.75 | -6.50 |
| Exposure to Screen Time in Hours | 4.54 ± 1.52 | 2.0 | 13.00 |
| Outdoor Activity in Minutes | 30.6 ± 11.34 | 15.0 | 120.00 |
| Tear Break Up Time in seconds | 10.34 ± 3.19 | 4 | 22 |
| Residual Astigmatism (DC) | -0.08 ± 1.35 | -2.50 | 1.75 |
| Near Point of Convergence (cm) | 7.41 ± 1.51 | 0.00 | 14.00 |
| Magnitude of Deviation (pd) | 6.27 ± 11.75 | 0 | 50 |
| Blink Per Minute | 10.07 ± 3.03 | 3 | 22 |
| Corneal Pachymetry | 542 ± 32.43 | 480 | 610 |

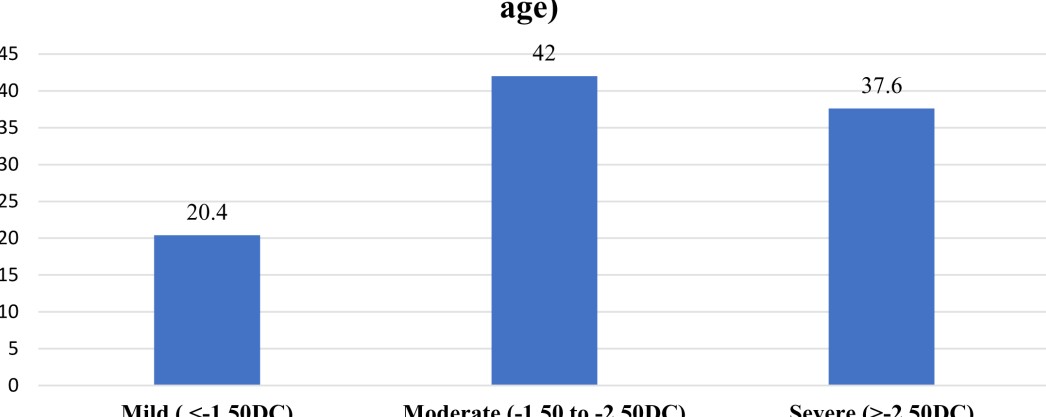

**Classification of Astigmatism on the Basis of Severity (% age)**

Mild ( <-1.50DC): 20.4
Moderate (-1.50 to -2.50DC): 42
Severe (>-2.50DC): 37.6

**Fig 1. Showed classification and magnitude of astigmatism based on severity.** Majority of the cylindrical prescription were lying in moderate and high form (Mild -0.75 to -1.50, Moderate> -1.50 to -2.50, High > -2.50).

severe in 0.9% of cases. The mean blink rate while using screens was 10.12 ± 3.0 blinks per minute, which is less than the average blink rate of 15-25 times per minute. Limited outdoor activities were reported by 87.67% of participants. Meridional amblyopia was found in 54.26% of the study participants.

The univariate analysis showed significant relationship among screen time exposure (independent variable), total astigmatism, and TBUT, while the results were insignificant for family history and gender analysis (Tables 2 and 3). The magnitude of total and corneal astigmatism showed a positive correlation with screen time (r = 0.33, p-value < 0.001; Table 4 ). This indicates that increasing screen time is associated with an increasing magnitude of astigmatism. The association of screen time and magnitude of astigmatism was further explored using regression analysis that showed a significant relationship (B = 0.177, p < 0.001; Table 3). It revealed that as the

Table 2 .  Univariate analysis among excessive screen time exposure and dependent variables.

| Screen time Exposure (Constant) | B | t value | p-Value | Conf. Interval |
|---|---|---|---|---|
| Total Astigmatism | 0.177 | 3.732 | <0.001 | 0.080-0.254 |
| TBUT | -0.154 | -3.233 | <0.001 | -0.50- -0.12 |
| Gender | 0.078 | 1.625 | 0.105 | -0.050-0.524 |
| Family History | 0.031 | 0.648 | 0.518 | -0.220-0.436 |

Table 3.  Effect of excessive screen exposure on total astigmatism, corneal astigmatism and TBUT.

| Screen time Exposure (Constant) | B | t value | p-Value | Conf. Interval |
|---|---|---|---|---|
| **Total Astigmatism and Corneal Astigmatism** | 0.177 | 3.732 | <0.001 | 0.080 to 0.254 |
| **TBUT** | -0.154 | -3.233 | <0.001 | -0.50 to -0.12 |

*Bonferroni Correction: 0.025.

Table 4.  The Spearman correlation analysis revealed a statistically significant negative correlation among exposure to screen time and TBUT while positive correlation with total astigmatism.

| Exposure to Screen time in Hours (Constant) | Correlation Coefficient (rho) | 95% Confidence Interval | p-Value |
|---|---|---|---|
| Total Astigmatism | 0.327 | 1.0 - 0.24 | <0.001 |
| Tear Break Up Time | -0.167 | -0.074 - -0.258 | |

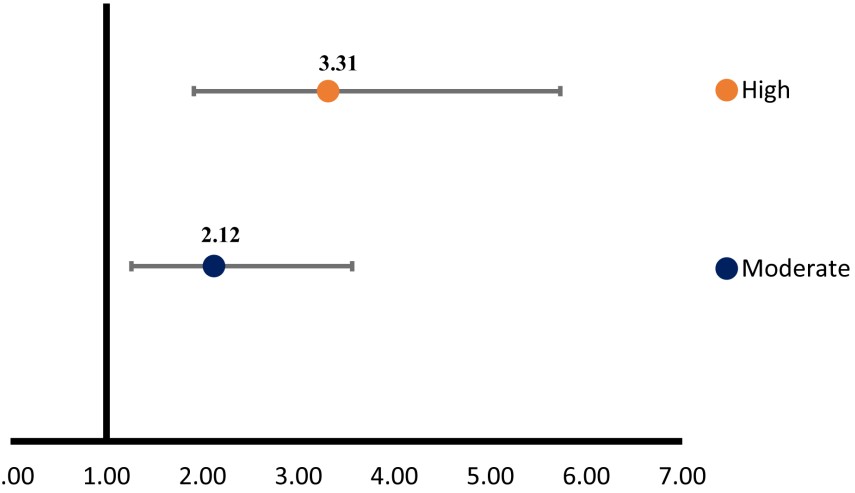

**Fig 2.  The forest plot representing the results of association between thickening of the upper eye lid due to inflammatory conjunctivitis in patients exposed to excessive screen time and astigmatism grades, i.e., mild, moderate and high.** The plot visually displays the strength and significance of the association, with the horizontal line representing the confidence interval and the dots marking the odds ratio. Since the confidence interval does not cross this line, it suggests a statistically significant association. (Ref value 1: Mild Astigmatism -0.75 to -1.50DC).

screen time increases, the magnitude of astigmatism also increases. The screen time and TBUT showed a significant negative correlation (r = -0.167, p-value < 0.001). However, the association between TBUT and inflammatory conjunctival tissue/lid thickness was significant ($\chi^2$ = 166.787; p < 0.001; Table 4), the regression analysis highlighted that unstable tear film (B = -0.431, p <

**Table 5. The results highlighted the odds ratio of inflammatory conjunctivitis related lid thickness and corneal astigmatism in patients exposed to excessive screen time.**

| Sr. No | Astigmatism | Odds ratio | 95% CI (Lower – Upper) |
|---|---|---|---|
| 1 | Low (-0.75 to -1.50) | Ref | |
| 2 | Moderate (>-1.50 to -2.50DC) | 2.12 | 1.26 – 3.56 |
| 3 | High (>-2.50DC) | 3.31 | 1.91 – 5.73 |

0.001; Table 3) is a risk factor for causing inflammatory conjunctivitis and lid thickening. The risk of inflammatory conjunctival tissue and lid thickness increases by 35% for each second of decrease in TBUT. Patients with inflammatory conjunctivitis/lid thickness had significantly higher odds of developing high astigmatism (OR = 3.31, p-value < 0.001, CI = 1.91 to 5.73) and moderate astigmatism (OR = 2.12, p-value = 0.004, CI = 1.26 to 3.56) Fig 2, Table 5). However, the effect of lid thickness on astigmatism when combining with screen time has a little effect that is not significant (p-value = 0.053). Thus, screen time is an independent risk factor of causing astigmatism in children (p < 0.001). The Wilcoxon Signed Ranks Test indicated a statistically significant improvement in visual acuity after correction. The visual acuity was significantly better in patients with correction (Z = 17.992; p-value < 0.001)

## Discussion

This study investigated the effect of excessive screen time exposure on visual and ocular health, specifically focusing on the relationships between excessive screen time, astigmatism, tear film stability (TBUT) and inflammatory conjunctivitis. With the rule astigmatism was observed in all children, and the most common type was mixed astigmatism. Moreover, there were factors associated with forceful blinking and lid thickening. Exotropia was more frequently observed in patients with uncorrected astigmatism

The risk of developing astigmatism potentially due to screen time exposure based on gender have contradictory reports and seems to vary in different populations. In this study we found that males were more prone to the effects of screen time that leads to astigmatism compared to female children. These findings are supported by studies from other populations [11,19]. Conversely, some studies have indicated that the prevalence of astigmatism is significantly higher among female students compared to their male counterparts while others have found no correlation between astigmatism and gender [20,21]. A major outcome of this study was the effect of excessive screen time on astigmatism. There are multiple factors associated with astigmatism, including environmental conditions, maternal/paternal smoking (active or passive), genetics, lifestyle, birth history, and feeding patterns. This study revealed that only 2.2% of the children were exposed to smoke during early childhood, suggesting no significant association between astigmatism and parental smoking. Also, most participants were breast-fed, as opposed to those who were bottle-fed or had mixed feeding patterns. The findings further indicated that, unlike previous studies, neither bottle feeding, nor c-sections had a significant impact on the development of astigmatism. [10,11,22,23].

A previously published study reported 33.1% prevalence of parental refractive error, while others found that children with a positive family history of astigmatism were more likely to develop astigmatism [6,8]. In comparison, this study observed a similar prevalence of parental refractive error at 25.5%, aligning with previously published data. These findings underscore the genetic predisposition to refraractive error specifically astigmatism, as demonstrated in prior research [8]. Moreover, the high education levels and annual incomes of the parents in current study may contribute to enhanced access to healthcare and preventive measures.

However, this accessibility may also increase risks, as excessive exposure to electronic devices among children might cause ocular discomfort and the onset of astigmatism.

Excessive screen time is considered a potential risk factor for developing astigmatism, either directly or indirectly. The study highlighted that there is a positive correlation among excessive screen time exposure and astigmatism ($p < 0.001$). The higher exposure to screen time is associated with a greater magnitude of astigmatism. A significant positive relationship between the amount of screen time in hours and the severity of astigmatism was found ($p \leq 0.001$). This suggested that for each additional hour of screen time, the magnitude of astigmatism will increase by 0.166 DC ($p < 0.001$). Excessive exposure to screens has been identified as a significant factor influencing vision development [24]. This may be linked to the impact on lens development, the low tensile strength of the cornea, pressure of upper lid on cornea in lower gaze and alterations in corneal shape due to prolonged exposure to smart screens [11,25]. This notion is supported by a study conducted in the Longhua district of Shenzhen among preschool children, where a significant association was found between early screen exposure and an elevated risk of astigmatism [10]. The risk of developing astigmatism was positively correlated with duration of screen exposure. Similar findings are also reported in other studies, further validating our results [26,27]. This study aligns with the previously published studies, showing a marked increase in astigmatism prevalence when daily screen time exceeds 2 hours, highlighting the importance of managing screen time to mitigate its impact on ocular health.

The results of excessive screen time exposure and TBUT revealed a negative correlation ($p < 0.001$; Table 4). This suggests that excessive screen time is associated with reduction in tear film stability. Specifically, for each additional hour of screen time, TBUT decreases by approximately 0.306 seconds. This suggests that higher screen time is associated with faster tear film breakup, potentially indicating a higher risk of dry eye symptoms or discomfort. Symptomatic DEDs associated with excessive screen time are linked to incomplete or partial blinking, which may lead to altered osmolarity and the buildup of inflammatory mediators. These inflammatory mediators are responsible for causing inflammatory conjunctivitis [28]. This study revealed a significant association between TBUT and inflammatory conjunctivitis/ lid thickness ($\chi^2 = 166.787$, $p < 0.001$). The results of TBUT and inflammatory conjunctivitis was significant ($p < 0.001$) supporting strong evidence of association between these variables that leads to thickening of the lid tarsal plate (Table 5). To explore the role of inflammatory conjunctivitis/lid thickness in the development of astigmatism, odds ratios were calculated (Fig 2). It showed that individuals with inflammatory conjunctivitis/lid thickening were significantly more likely to develop astigmatism, supporting the theory of mechanical reshaping of the cornea. However, the multinomial regression analysis of astigmatism and thickened tarsal conjunctival plate in relation to screen time revealed that, although lid thickness is associated with astigmatism, its impact is minor. In contrast, screen time emerged as an independent and significant risk factor for astigmatism development. The study also showed that not every child has a thicker lid due to inflammatory conjunctivitis. Prolonged lower gaze while staring at a screen itself has the potential to reshape the cornea, as its tensile strength is less in young individuals. The physiological link between screen exposure and astigmatism remains unclear, though several hypotheses exist. One possibility is that prolonged screen time at close distances may lead to excessive accommodation, which overworks the ciliary muscles in children's eyes and potentially disrupts the natural development of the crystalline lens, altering its curvature [24]. Another hypothesis suggests that close-up screen viewing could change the shape of children's corneas due to variations in the palpebral aperture and eye movements during these tasks or due to mechanical interactions between the cornea and eyelids [29]. Additionally, it has been proposed that looking down at a screen may increase pressure from

the eyelids on the cornea, potentially leading to steepening of the vertical corneal meridian, causing with-the-rule astigmatism [25,30] Other findings of the study was presence of exotropia mostly intermittent that were common among children exposed to screen time and astigmatism. Astigmatic blur is one of the reasons that can contribute to the disruption of binocular vision, particularly in individuals with weak fusional control, potentially leading to the development of exotropia. A recently published studies by Çakır B et al. and Shah M et al. revealed that intermittent exotropia is more common in anisoastigmatic children [16,31].

### Limitation of the study

This cross-sectional study highlights the association and risk factors of excessive screen time exposure with changes in ocular dimensions, particularly astigmatism, in children. While we have made every effort to minimize biases related to screen time exposure, it remains a limitation of the study.

### Conclusion

The current study highlighted the significant impact of excessive screen time on various aspects of visual and ocular health in children. The important aspects were the development of astigmatism, tear film instability, and associated conditions such as inflammatory conjunctivitis and exotropia. The study also revealed the adverse effect of screen time on TBUT, contributing to digital eye strain and dry eye symptoms. Furthermore, the research suggests that inflammatory conjunctivitis and lid thickening, exacerbated by screen time, may play a role in the mechanical reshaping of the cornea, leading to astigmatism. These results emphasize the need for preventive measures, such as limiting screen time and promoting regular eye examinations, to mitigate the adverse effects of smart screens on children's ocular health.

### Supporting information

**S1 Fig.  The figure highlights screen time usage among children with dedicated devices**.
(PDF)

**S2 Fig.  The figure showed the graphical presentation of the histogram plots highlighting the normality of the data.**
(PDF)

**S3 Fig.  This figure showed parents educational level of the study participants.** It showed that mostly they were educated to graduate or post graduate level.
(PDF)

**S4 Fig.  Presented the status of parental income status per annum as per Govt of Pakistan Tax Slab status, Majority were lying in tab slab 3 and 4.**
(PDF)

**S5 Fig.  Represented the type of Astigmatism on the basis of Foci, it showed that majority of the study participants were diagnosed with Mixed Astigmatism followed by Myopic and Hypermetropic Astigmatism.**
(PDF)

**S1 File.  Inclusivity in global research questionnaire.**
(DOCX)

**S2 File.  SPSS data.** File of Association of Escessive Screen time exposure in children.
(SAV)

## Acknowledgments

I sincerely thank my HOD, Dr. Sufian Ali Khan, for his invaluable guidance, and my colleagues, Hashim Ali Khan, Saif Ullah, Muhammad Zubair, and Saad Alam Khan, for their continuous support throughout this study.

## Author contributions

**Conceptualization:** Mutahir Shah, Satheesh Babu Natarajan, Nafees Ahmad.

**Data curation:** Mutahir Shah, Nafees Ahmad.

**Formal analysis:** Mutahir Shah, Satheesh Babu Natarajan, Nafees Ahmad.

**Investigation:** Mutahir Shah, Nafees Ahmad.

**Methodology:** Mutahir Shah, Satheesh Babu Natarajan, Nafees Ahmad.

**Project administration:** Satheesh Babu Natarajan.

**Software:** Satheesh Babu Natarajan, Nafees Ahmad.

**Supervision:** Satheesh Babu Natarajan, Nafees Ahmad.

**Visualization:** Nafees Ahmad.

**Writing – original draft:** Mutahir Shah.

**Writing – review & editing:** Nafees Ahmad.

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
