## [Decision Letter · Decision Letter 0]

31 Oct 2024

PONE-D-24-41901

Association of excessive screen time exposure with ocular changes leading to astigmatism in children

PLOS ONE

Dear Dr. Shah,

Thank you for submitting your manuscript to PLOS ONE. After careful consideration, we feel that it has merit but does not fully meet PLOS ONE’s publication criteria as it currently stands. Therefore, we invite you to submit a revised version of the manuscript that addresses the points raised during the review process.

We look forward to receiving your revised manuscript.

Kind regards,

Clara Martínez Pérez

Academic Editor

PLOS ONE

Reviewers' comments:

Reviewer's Responses to Questions

**Comments to the Author**

1. Is the manuscript technically sound, and do the data support the conclusions?

Reviewer #1: Partly

Reviewer #2: Partly

2. Has the statistical analysis been performed appropriately and rigorously? 

Reviewer #1: Yes

Reviewer #2: No

3. Have the authors made all data underlying the findings in their manuscript fully available?

Reviewer #1: No

Reviewer #2: No

4. Is the manuscript presented in an intelligible fashion and written in standard English?

Reviewer #1: Yes

Reviewer #2: Yes

5. Review Comments to the Author

Reviewer #1: I acknowledge the hard work done by the author. The paper doesnot carry any novelty. This is a simple study with the inference which is already known as a fact.

The authors haven't ruled out various other factors responsible for early onset keratoconus.

The exclusion criteria hasnot mentioned the vernal keratoconjunctivitis

Above all, how authors have negated the effect of the hereditary astigmatism as they have mentioned positive family history also.

The lid thickness has been mentioned in the manuscript time and again, how it was measured is not mentioned anywhere,

The astigmatism mentioned can be just a result of various other factors. It could have been better if the corneal thickness have been mentioned for these patients.

Reviewer #2: This study investigates the association between excessive screen time and ocular changes, specifically focusing on astigmatism, tear film instability, and inflammatory conditions in children. Conducted on 431 participants, it uses various statistical tests to examine relationships between screen time and astigmatism, tear break-up time (TBUT), and related ocular inflammation.

1. The accuracy of screen time measurement requires clarification, given its reliance on parental reporting or device records, which may not exclusively reflect children’s usage. Additionally, clarification is needed on whether television viewing was included as part of the screen time metrics, as it may affect the overall exposure assessment.

2. The classification of this study as a cross-sectional correlational study may be inaccurate, as it comprises a single group sharing the same outcome. This design more closely resembles a case series, which is essential to clarify to align with accurate study categorization.

3. The manuscript states an initial sample of 431 cases; however, the subgroup totals (84 + 175 + 157 = 416) do not sum to this figure. A flow diagram detailing any exclusions or attrition from sampling to the final analysis would enhance transparency.

4. Given the large sample size, the Shapiro-Wilk test might yield statistically significant results without meaningful deviations. To address this, normality should be evaluated via central tendency and dispersion measures alongside visual inspections against normal distributions.

5. Due to multiple comparisons, the standard significance level of 0.05 should be adjusted (e.g., Bonferroni correction) to control for the increased risk of Type I errors.

6. Visual acuity, as a fractional index, would be more accurately represented using the harmonic mean instead of the arithmetic mean, aligning with statistical best practices for fractional data.

7. The exclusion of key variables, such as gender and family history, from the multivariable analysis raises concerns about unaccounted confounding/covariate effects. Including these in stepwise multivariable regression could provide more robust estimates of variable impact.

8. Reporting the 95% confidence intervals for correlation coefficients (r) and including interpretation ranges for "r" and "standardized beta" values would aid readers in evaluating the strength of associations.

9. A table summarizing the data used to derive the odds ratios (OR) would enhance clarity and allow for independent verification of these estimates.

10. The manuscript should present all diagnostic assessments for linear regression assumptions, such as normality of residuals, homoscedasticity, and the presence of outliers or leverage points. Additionally, any collinearity diagnostics and how these were managed would improve the methodological rigor.

6. PLOS authors have the option to publish the peer review history of their article (what does this mean? ). If published, this will include your full peer review and any attached files.

**Do you want your identity to be public for this peer review?** For information about this choice, including consent withdrawal, please see our Privacy Policy .

Reviewer #1: No

Reviewer #2: No

---

## [Author Response · Author response to Decision Letter 1]

4 Dec 2024

Rebuttal Letter

Reviewer 1

Q1. The authors haven't ruled out various other factors responsible for early onset keratoconus?

Ans: The study highlighted the association and effect of excessive screen time exposure, ocular dimension change and astigmatism development. We excluded all those factors associated with early onset of keratoconus including risk factors, clinical signs, corneal topography parameters based on Rabinowitz criteria and corneal thickness. All those factors that were associated or may be a risk factor of developing keratoconus in future were excluded. (P. No 06, Line No 110-111)

Q2. The exclusion criteria have not mentioned the vernal keratoconjunctivitis?

Ans: Indeed, it was necessary to mention atopy, vernal keratoconjunctivitis, and syndromes associated with keratoconus. These factors have now been included in the exclusion criteria, as suggested by the esteemed reviewer. (P. No 06, Line No 110-111)

Q3. how authors have negated the effect of the hereditary astigmatism as they have mentioned positive family history?

Ans: A univariate analysis was performed to rule out the effects of hereditary astigmatism, but the results were insignificant. This has been updated in the main manuscript, both in the methodology and results sections. (P.no. 7, 10, 11, Line No.138,139,184-86, 213-14)

Q4. The lid thickness has been mentioned in the manuscript time and again, how it was measured is not mentioned anywhere?

Ans: Excessive screen time is negatively associated with TBUT resulting in tear film instability responsible for Inflammatory conjunctivitis. The inflammatory conjunctivitis was supposed to be associated with thickening of the tarsal plat of the conjunctiva shown in literature. The thickening or swelling of the conjunctival tarsal plate was clinically assessed on slit lamp examination based on conjunctival inflammation and digital assessment for overall lid rigidity. The study statistically highlighted that this rigidity of lid is influencing corneal curvature thus causing with the rule astigmatism. (P. no 7, Line No 128-131)

Q5. It could have been better if the corneal thickness has been mentioned for these patients?

Ans: The corneal thickness in main table as well as in result section has mentioned and updated. (P. no: 8,9, Line No 165-66, 174)

Reviewer 2

Q1: The accuracy of screen time measurement requires clarification, given its reliance on parental reporting or device records, which may not exclusively reflect children’s usage. Additionally, clarification is needed on whether television viewing was included as part of the screen time metrics, as it may affect the overall exposure assessment.

Ans: Screen time was evaluated using smartphone usage history, supported by parental reports for children with dedicated gadgets. For non-dedicated gadgets, parental reports on their children's usage were utilized (Fig S5). While we have made every effort to minimize biases related to screen time exposure, it remains a limitation of the study. Exposure to TV screens was excluded from the study because recent research predominantly associates it with myopia rather than astigmatism. (P. No 6, Line No114-17)

Q2. The classification of this study as a cross-sectional correlational study may be inaccurate, as it comprises a single group sharing the same outcome. This design more closely resembles a case series, which is essential to clarify to align with accurate study categorization?

Ans: The esteemed reviewer highlighted a significant point. While the cases bear similarities, the large dataset makes it more suitable for a cross-sectional study, focusing on identifying the relationship between excessive screen time exposure and astigmatism.

Q3. The manuscript states an initial sample of 431 cases; however, the subgroup totals (84 + 175 + 157 = 416) do not sum to this figure. A flow diagram detailing any exclusions or attrition from sampling to the final analysis would enhance transparency?

Ans: This was both a typographical and calculation error. The corrections have been made in the results section as well as in the graphical representation. The updated study now includes a total of 431 cases. (P no. 8, Line No 163-64)

Q 4. Given the large sample size, the Shapiro-Wilk test might yield statistically significant results without meaningful deviations. To address this, normality should be evaluated via central tendency and dispersion measures alongside visual inspections against normal distributions.

Ans: As mentioned, the data did not follow a normal distribution based on the Shapiro-Wilk test, which was further confirmed through normality assessment using histogram plots (Figure S4). These updates have been incorporated into both the methodology and results sections (P no. 7, Lines 135-36).

Q5. Due to multiple comparisons, the standard significance level of 0.05 should be adjusted (e.g., Bonferroni correction) to control for the increased risk of Type I errors.

Ans: As suggested by the esteemed reviewer, the Bonferroni adjustment for multiple comparisons was applied to establish the p-value for the multivariable analysis, which was calculated to be 0.025. The results of the study remained significant, effectively controlling the risk of Type I errors (Page no 7, Line No:141,142, 216).

Q6. Visual acuity, as a fractional index, would be more accurately represented using the harmonic mean instead of the arithmetic mean, aligning with statistical best practices for fractional data.

Ans: We sincerely appreciate the valuable suggestions provided by the esteemed reviewer. We chose to use arithmetic means instead of harmonic means because the corrected visual acuity (VA) dataset includes a value of 0. In the LogMAR (Logarithm of the Minimum Angle of Resolution) scale, a value of 0 corresponds to a visual acuity of 6/6, which represents standard vision. The inclusion of 0 makes it impossible to calculate harmonic means, as the harmonic mean is undefined in the presence of zero values. While we included the harmonic mean for uncorrected VA, we excluded it for corrected VA for this reason. Visual acuity is commonly expressed using four major notations: Feet, Meters, Decimal, and LogMAR. Among these, the LogMAR scale is widely regarded as the gold standard for research studies due to its precision. The scale assigns specific values to each letter on the chart and is structured in fractions, offering greater accuracy for statistical analysis and cross-study comparisons. This approach ensures that even subtle changes in visual acuity are captured reliably. (P. N0. 8, Line No. 158, 59)

Q7. The exclusion of key variables, such as gender and family history, from the multivariable analysis raises concerns about unaccounted confounding/covariate effects. Including these in stepwise multivariable regression could provide more robust estimates of variable impact.

Ans: The study included univariate analysis to identify associations and significance. Gender and family history showed insignificant results, indicating their negligible role in the overall study. Therefore, they were not included in the multivariate analysis. These points have been discussed in the methodology section and updated in the results section (P no. 7, Line No: 138-39)

Q8. Reporting the 95% confidence intervals for correlation coefficients (r) and including interpretation ranges for "r" and "standardized beta" values would aid readers in evaluating the strength of associations.

Ans. As suggested by the reviewer, the confidence interval levels for correlation coefficients and beta values have been included to provide greater clarity for readers in evaluating the strength of associations. (P no 11, Line No.211-216)

Q9. A table summarizing the data used to derive the odds ratios (OR) would enhance clarity and allow for independent verification of these estimates.

Ans: A table has been added, along with the forest plot, to enhance clarity and allow for independent verification of these estimates. (P. No. 12, Line No. 217-18

Q10. The manuscript should present all diagnostic assessments for linear regression assumptions, such as normality of residuals, homoscedasticity, and the presence of outliers or leverage points. Additionally, any collinearity diagnostics and how these were managed would improve the methodological rigor.

Ans. The outliers and leverage points were not significant in this study, so no further analysis was conducted.

---

## [Decision Letter · Decision Letter 1]

16 Dec 2024

PONE-D-24-41901R1Association of excessive screen time exposure with ocular changes leading to astigmatism in childrenPLOS ONE

Dear Dr. Shah,

Thank you for submitting your manuscript to PLOS ONE. After careful consideration, we feel that it has merit but does not fully meet PLOS ONE’s publication criteria as it currently stands. Therefore, we invite you to submit a revised version of the manuscript that addresses the points raised during the review process.

We look forward to receiving your revised manuscript.

Kind regards,

Clara Martínez Pérez

Academic Editor

PLOS ONE

Journal Requirements:

Comments from PLOS Editorial Office: In addition to addressing the concerns raised by the reviewers, please could you also address the following issues:

*1) The Table 2 caption reads "The Spearman correlation analysis revealed a statistically significant negative correlation between exposure to screen time, TBUT and, total and corneal astigmatism." However, two of the 3 coefficients presented in the table are positive, not negative.*

*2) There appears to be a conflation between odds and risk ratios. You report that "inflammatory conjunctivitis/lid*

*thickness have three times greater risk of developing high astigmatism (OR = 3.31, p-value < 0.001, CI = 1.91 to 5.73) while the risk of moderate astigmatism in such cases was two times higher (2.12, p-value = 0.004, CI=.26-3.56)." However, an odds ratio of 3 is not the same as a three-fold increase in risk.*

Reviewers' comments:

Reviewer's Responses to Questions

**Comments to the Author**

1. If the authors have adequately addressed your comments raised in a previous round of review and you feel that this manuscript is now acceptable for publication, you may indicate that here to bypass the “Comments to the Author” section, enter your conflict of interest statement in the “Confidential to Editor” section, and submit your "Accept" recommendation.

Reviewer #1: All comments have been addressed

Reviewer #2: All comments have been addressed

2. Is the manuscript technically sound, and do the data support the conclusions?

Reviewer #1: Yes

Reviewer #2: Yes

3. Has the statistical analysis been performed appropriately and rigorously? 

Reviewer #1: Yes

Reviewer #2: Yes

4. Have the authors made all data underlying the findings in their manuscript fully available?

Reviewer #1: Yes

Reviewer #2: Yes

5. Is the manuscript presented in an intelligible fashion and written in standard English?

Reviewer #1: Yes

Reviewer #2: Yes

6. Review Comments to the Author

Reviewer #1: (No Response)

Reviewer #2: While the authors have adequately addressed the majority of the comments, I have a concern regarding the sample size calculation. It appears that the formula used is appropriate for descriptive studies; however, given the nature of the study, a formula specific to correlational studies would be more suitable. Therefore, I request the authors to perform a post-hoc power analysis to confirm whether the sample size is sufficient to support the study's conclusions.

7. PLOS authors have the option to publish the peer review history of their article (what does this mean? ). If published, this will include your full peer review and any attached files.

**Do you want your identity to be public for this peer review?** For information about this choice, including consent withdrawal, please see our Privacy Policy .

Reviewer #1: No

Reviewer #2: No

---

## [Author Response · Author response to Decision Letter 2]

23 Dec 2024

Thank you for your valuable comments. Regarding the conflation of odds ratios and risk ratios, we acknowledge the distinction and have revised our phrasing to accurately report odds ratios (ORs) without implying equivalence to relative risks. For the sample size concern, we conducted a post-hoc power analysis using G*Power software, which confirmed that a sample size of 100 would have been sufficient; with our actual sample size of 431, the study is adequately powered. We appreciate your feedback, which has helped us strengthen the clarity and rigor of our work.

---

## [Decision Letter · Decision Letter 2]

8 Jan 2025

Association of excessive screen time exposure with ocular changes leading to astigmatism in children

PONE-D-24-41901R2

Dear Dr. Shah,

We’re pleased to inform you that your manuscript has been judged scientifically suitable for publication and will be formally accepted for publication once it meets all outstanding technical requirements.

Kind regards,

Clara Martínez Pérez

Academic Editor

PLOS ONE

Additional Editor Comments (optional):

Reviewers' comments:

Reviewer's Responses to Questions

**Comments to the Author**

1. If the authors have adequately addressed your comments raised in a previous round of review and you feel that this manuscript is now acceptable for publication, you may indicate that here to bypass the “Comments to the Author” section, enter your conflict of interest statement in the “Confidential to Editor” section, and submit your "Accept" recommendation.

Reviewer #1: All comments have been addressed

Reviewer #2: All comments have been addressed

2. Is the manuscript technically sound, and do the data support the conclusions?

Reviewer #1: Yes

Reviewer #2: Yes

3. Has the statistical analysis been performed appropriately and rigorously? 

Reviewer #1: Yes

Reviewer #2: Yes

4. Have the authors made all data underlying the findings in their manuscript fully available?

Reviewer #1: Yes

Reviewer #2: Yes

5. Is the manuscript presented in an intelligible fashion and written in standard English?

Reviewer #1: Yes

Reviewer #2: Yes

6. Review Comments to the Author

Reviewer #1: Exclusion criteria needed revision which is done. Astigmatism is a combination of various ocular factors which contributes to it and needs to be addressed in the manuscript.

Reviewer #2: I appreciate the authors efforts; all comments are appropriately addressed and no more comments or revisions are needed

7. PLOS authors have the option to publish the peer review history of their article (what does this mean? ). If published, this will include your full peer review and any attached files.

**Do you want your identity to be public for this peer review?** For information about this choice, including consent withdrawal, please see our Privacy Policy .

Reviewer #1: **Yes: ** Prof Dr Anuradha Raj

Reviewer #2: No

---

## [Editor Report · Acceptance letter]

PONE-D-24-41901R2

PLOS ONE

Dear Dr. Shah,

I'm pleased to inform you that your manuscript has been deemed suitable for publication in PLOS ONE. Congratulations! Your manuscript is now being handed over to our production team.

Kind regards,

on behalf of

Dr. Clara Martínez Pérez

Academic Editor

PLOS ONE